# Uncertainty-Aware Attention for Reliable Interpretation and Prediction

**Jay Heo**[1,2,4*]**, Hae Beom Lee**[1,2*]**, Saehoon Kim**[2]**, Juho Lee**[2,5]**, Kwang Joon Kim**[3]**,**
**Eunho Yang**[1,2]**, Sung Ju Hwang**[1,2]

KAIST[1], AItrics[2], Yonsei University College of Medicine[3], UNIST[4], South Korea,
University of Oxford[5], United Kingdom,
{jayheo, haebeom.lee, sjhwang82, eunhoy}@kaist.ac.kr
shkim@aitrics.com, preppie@yuhs.ac, juho.lee@stats.ox.ac.uk

## Abstract

Attention mechanism is effective in both focusing the deep learning models on relevant features and interpreting them. However, attentions may be unreliable since the networks that generate them are often trained in a weakly-supervised manner. To overcome this limitation, we introduce the notion of *input-dependent uncertainty* to the attention mechanism, such that it generates attention for each feature with varying degrees of noise based on the given input, to learn larger variance on instances it is uncertain about. We learn this *Uncertainty-aware Attention (UA)* mechanism using variational inference, and validate it on various risk prediction tasks from electronic health records on which our model significantly outperforms existing attention models. The analysis of the learned attentions shows that our model generates attentions that comply with clinicians' interpretation, and provide richer interpretation via learned variance. Further evaluation of both the accuracy of the uncertainty calibration and the prediction performance with "I don't know" decision show that UA yields networks with high reliability as well.

## 1 Introduction

For many real-world safety-critical tasks, achieving high reliablity may be the most important objective when learning predictive models for them, since incorrect predictions could potentially lead to severe consequences. For instance, failure to correctly predict the sepsis risk of a patient in ICU may cost his/her life. Deep learning models, while having achieved impressive performances on multitudes of real-world tasks such as visual recognition [17, 10], machine translation [2] and risk prediction for healthcare [3, 4], may be still susceptible to such critical mistakes since most do not have any notion of predictive uncertainty, often leading to overconfident models [9, 18] that are prone to making mistakes. Even worse, they are very difficult to analyze, due to multiple layers of non-linear transformations that involves large number of parameters.

Attention mechanism [2] is an effective means of guiding the model to focus on a partial set of most relevant features for each input instance. It works by generating (often sparse) coefficients for the given features in an input-adaptive manner, to allocate more weights to the features that are found to be relevant for the given input. Attention mechanism has been shown to significantly improve the model performance for machine translation [2] and image annotation [28] tasks. Another important feature of the attention mechanism is that it allows easy interpretation of the model via the generated attention allocations, and one recent work on healthcare domain [3] is focusing on this aspect.

---

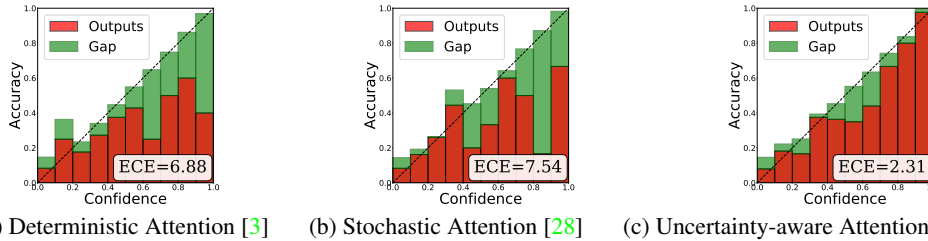

(a) Deterministic Attention [3]   (b) Stochastic Attention [28]   (c) Uncertainty-aware Attention (Ours)

Figure 1: Reliability diagrams [9] which shows the accuracy as a function of model confidence, generated from RNNs trained for mortality risk analysis from ICU records (PhysioNet-Mortality). ECE [22] in (8) denotes Expected Calibration Error, which is the weighted-average gap between model confidence and actual accuracy. (Gap is shown in green bars.) Conventional attention models result in poorly calibrated networks while our UA yields a well-calibrated one. Such accurately calibrated networks allow us to perform reliable prediction by leveraging prediction confidence to decide whether to predict or defer prediction.

Although interpretable, attention mechanisms are still limited as means of implementing safe deep learning models for safety-critical tasks, as they are not necessarily reliable. The attention strengths are commonly generated from a model that is trained in a weakly-supervised manner, and could be incorrectly allocated; thus they may not be safe to base final prediction on. To build a reliable model that can prevent itself from making critical mistakes, we need a model that knows its own limitation - *when it is safe to make predictions and when it is not*. However, existing attention model cannot handle this issue as they do not have any notion of predictive uncertainty. This problem is less of an issue in the conventional use of attention mechanisms, such as machine translation or image annotation, where we can often find clear link between the attended parts and the generated output. However, when working with variables that are often noisy and may not be one-to-one matched with the prediction, such as in case of risk predictions with electronic health records, the overconfident and inaccurate attentions can lead to incorrect predictions (See Figure 1).

To tackle this limitation of conventional attention mechanisms, we propose to allow the attention model to output uncertainty on each feature (or input) and further leverage them when making final predictions. Specifically, we model the attention weights as Gaussian distribution with *input-dependent noise*, such that the model generates attentions with small variance when it is confident about the contribution of the given features, and allocates noisy attentions with large variance to uncertain features, for each input. This input-adaptive noise can model *heteroscedastic uncertainty* [14] that varies based on the instance, which in turn results in uncertainty-based attenuation of attention strength. We formulate this novel uncertainty-aware attention (UA) model under the Bayesian framework and solve it with variational inference[2].

We validate UA on tasks such as sepsis prediction in ICU and disease risk prediction from electronic health records (EHR) that have large degree of uncertainties in the input, on which our model outperforms the baseline attention models by large margins. Further quantitative and qualitative analysis of the learned attentions and their uncertainties show that our model can also provide richer interpretations that align well with the clinician's interpretations. For further validation on prediction reliability, we evaluate it for the uncertainty calibration performance, and prediction under the scenario where the model can defer the decision by saying "I don't know", whose results show that UA yields significantly better calibrated networks that can better avoid making incorrect predictions on instances that it is uncertain, compared to baseline attention models.

Our contribution in this paper is threefold:

- We propose a novel variational attention model with instance-dependent modeling of variance, that captures input-level uncertainty and use it to attenuate attention strengths.

- We show that our uncertainty-aware attention yields accurate calibration of model uncertainty as well as attentions that aligns well with human interpretations.

- We validate our model on six real-world risk prediction problems in healthcare domains, for both the original binary classification task and classification with "I don't know" decision, and show that our model obtains significant improvements over existing attention models.

## 2 Related Work

**Prediction reliability** There has been work on building a *reliable* deep learning model[29, 13, 14]; that is, a deep network that can avoid making incorrect predictions when it is not sufficiently certain about its prediction. To achieve this goal, a model should know the limitation in the data, and in itself. One way to quantify such limitations is by measuring the *predictive uncertainty* using Bayesian models. Recently, [7, 5, 6] showed that deep networks with dropout sampling [24] can be understood as Bayesian neural networks. To obtain better calibrated dropout uncertainties, [15, 8] proposed to automatically learn the dropout rates with proper reparameterization tricks [21, 16]. While the aforementioned work mostly focus on accurate calibration of uncertainty itself, Kendall and Gal [14] utilized dropout sampling to model predictive uncertainty in computer vision [13, 26], and also modeled label noise with learned variances, to implicitly attenuate loss for the highly uncertain instances. Our work has similar motivation, but we model the uncertainty in the *input data* rather than in labels. By doing so, we can accurately calibrate deep networks for improved reliability. Ayhan et al. [1] has a similar motivation to ours, but with different applications and approaches. There exists quite a few work about uncertainty calibration and its quantification. Guo et al. [9] showed that the modern deep networks are poorly calibrated despite their accuracies, and proposed to tune factors such as depth, width, weight decay for better calibration of the model, and Lakshminarayanan et al. [18] proposed ensemble and adversarial training for the same objective.

**Attention mechanism** The literature on the attention mechanism is vast, which includes its application to machine translation [2], memory-augmented networks [25], and for image annotation [28]. Attention mechanisms are also used for interpretability, as in Choi et al. [3] which proposed a RNN-based attention generator for EHR that can provide attention on both the hospital visits and variables for further analysis by clincians. Attentions can be either deterministic or probabilistic, and soft (non-sparse) or hard (sparse). Some probabilistic attention models [28] use variational inference as used in our model. However, while their direct learning of multinoulli distribution only considers whether to attend or not without consideration of variance, our attention mechanism models varying degree of uncertainty for each input by input-dependent learning of attention noise (variance).

## 3 Approach

We now describe our uncertainty-aware attention model. Let $\mathcal{D}$ be a dataset containing a set of $N$ input data points $\mathbf{X} = [\mathbf{x}^{(1)} \dots \mathbf{x}^{(N)}]$ and the corresponding labels, $\mathbf{Y} = [\mathbf{y}^{(1)} \dots \mathbf{y}^{(N)}]$. For notational simplicity, we suppress the data index $n = 1, \dots, N$ when it is clear from the context.

We first present a general framework of a stochastic attention mechanism. Let $\mathbf{v}(\mathbf{x}) \in \mathbb{R}^{r \times i}$ be the concatenation of $i$ intermediate features, each column of which $\mathbf{v}_j(\mathbf{x})$ is a length $r$ vector, from an arbitrary neural network. From $\mathbf{v}(\mathbf{x})$, a set of random variables $\{\mathbf{a}_j\}_{j=1}^i$ is conditionally generated from some distribution $p(\mathbf{a}|\mathbf{x})$ where the dimension of $\mathbf{a}_j$ depends on the model architecture. Then, the context vector $\mathbf{c} \in \mathbb{R}^r$ is computed as $\mathbf{c}(\mathbf{x}) = \sum_{j=1}^i \mathbf{a}_j \odot \mathbf{v}_j(\mathbf{x})$ where the operator $\odot$ is properly defined according to the dimensionality of $\mathbf{a}_j$; if $\mathbf{a}_j$ is a scalar, it is simply the multiplication while for $\mathbf{a}_j \in \mathbb{R}^r$, it is the element-wise product. The function $f$ here produces the prediction $\hat{\mathbf{y}} = f(\mathbf{c}(\mathbf{x}))$ given the context vector $\mathbf{c}$.

The attention could be generated either deterministically, or stochastically. The stochastic attention mechanism is proposed in [28], where they generate $\mathbf{a}_j \in \{0, 1\}$ from Bernoulli distribution. This variable is learned by maximizing the evidence lower bound (ELBO) with additional regularizations for reducing variance of gradients. In [28], the stochastic attention is shown to perform better than the deterministic counterpart, on image annotation task.

### 3.1 Stochastic attention with input-adaptive Gaussian noise

Despite the performance improvement in [28], there are two limitations in modeling stochastic attention directly with Bernoulli (or Multinoulli) distribution as [28] does, in our purposes:

**1) The variance $\sigma^2$ of Bernoulli is completely dependent on the allocation probability $\mu$.**

Since the variance for Bernoulli distribution is decided as $\sigma^2 = \mu(1 - \mu)$, the model thus cannot generate $\mathbf{a}$ with low variance if $\mu$ is around $0.5$, and vice versa. To overcome such limitation, we

disentangle the attention strength $\mathbf{a}$ from the attention uncertainty so that the uncertainty could vary even with the same attention strength.

**2) The vanilla stochastic attention models the noise independently of the input.**

This makes it infeasible to model the amount of uncertainty for each input, which is a crucial factor for reliable machine learning. Even for the same prediction tasks and for the same set of features, the amount of uncertainty for each feature may largely vary across different instances.

To overcome these two limitations, we model the standard deviation $\sigma$, which is indicative of the uncertainty, as an *input-adaptive* function $\sigma(\mathbf{x})$, enabling to reflect different amount of confidence the model has for each feature, for a given instance. As for distribution, we use *Gaussian* distribution, which is probably the most simple and efficient solution for our purpose, and also easy to implement.

We first assume that a subset of the neural network parameters $\boldsymbol{\omega}$, associated with generating attentions, has zero-mean isotropic Gaussian prior with precision $\tau$. Then the attention scores before squashing, denoted as $\mathbf{z}$, are generated from conditional distribution $p_\theta(\mathbf{z}|\mathbf{x}, \boldsymbol{\omega})$, which is also Gaussian:

$$p(\boldsymbol{\omega}) = \mathcal{N}(\mathbf{0}, \tau^{-1}\mathbf{I}), \quad p_\theta(\mathbf{z}|\mathbf{x}, \boldsymbol{\omega}) = \mathcal{N}(\boldsymbol{\mu}(\mathbf{x}, \boldsymbol{\omega}; \theta), \operatorname{diag}(\boldsymbol{\sigma}^2(\mathbf{x}, \boldsymbol{\omega}; \theta))) \tag{1}$$

where $\boldsymbol{\mu}(\cdot, \boldsymbol{\omega}; \theta)$ and $\boldsymbol{\sigma}(\cdot, \boldsymbol{\omega}; \theta)$ are mean and s.d., parameterized by $\theta$. Note that $\boldsymbol{\mu}$ and $\boldsymbol{\sigma}$ are generated from the same layer, but with different set of parameters, although we denote those parameters as $\theta$ in general. The actual attention $\mathbf{a}$ is then obtained by applying some squashing function $\pi(\cdot)$ to $\mathbf{z}$ (e.g. sigmoid or hyperbolic tangent): $\mathbf{a} = \pi(\mathbf{z})$. For comparison, one can think of the vanilla stochastic attention of which variance is independent of inputs.

$$p(\boldsymbol{\omega}) = \mathcal{N}(\mathbf{0}, \tau^{-1}\mathbf{I}), \quad p_\theta(\mathbf{z}|\mathbf{x}, \boldsymbol{\omega}) = \mathcal{N}(\boldsymbol{\mu}(\mathbf{x}, \boldsymbol{\omega}; \theta), \operatorname{diag}(\boldsymbol{\sigma}^2)) \tag{2}$$

However, as we mentioned, this model cannot express different amount of uncertainties over features.

One important aspect of our model is that, in terms of graphical representation, the distribution $p(\boldsymbol{\omega})$ is independent of $\mathbf{x}$, while the distribution $p_\theta(\mathbf{z}|\mathbf{x}, \boldsymbol{\omega})$ is conditional on $\mathbf{x}$. That is, $p(\boldsymbol{\omega})$ tends to capture uncertainty of model parameters (epistemic uncertainty), while $p_\theta(\mathbf{z}|\mathbf{x}, \boldsymbol{\omega})$ reacts sensitively to uncertainty in data, varying across different input points (heteroscedastic uncertainty) [14]. When *modeled together*, it has been empirically shown that the quality of uncertainty improves [14]. Such modeling both input-agnostic and input-dependent uncertainty is especially important in risk analysis tasks in healthcare, to capture both the uncertainty from insufficient amount of clinical data (e.g. rare diseases), and the uncertainty that varies from patients to patients (e.g. sepsis).

## 3.2  Variational inference

We now model what we have discussed so far. Let $\mathbf{Z}$ be the set of latent variables $\{\mathbf{z}^{(n)}\}_{n=1}^N$ that stands for attention weight before squashing. In neural network, the posterior distribution $p(\mathbf{Z}, \boldsymbol{\omega}|\mathcal{D})$ is usually computationally intractable since $p(\mathcal{D})$ is so due to nonlinear dependency between variables. Thus, we utilize variational inference, which is an approximation method that has been shown to be successful in many applications of neural networks [16, 23], along with reprameterization tricks for pathwise backpropagation [15, 8].

Toward this, we first define our variational distribution as

$$q(\mathbf{Z}, \boldsymbol{\omega}|\mathcal{D}) = q_{\mathbf{M}}(\boldsymbol{\omega}|\mathbf{X}, \mathbf{Y})q(\mathbf{Z}|\mathbf{X}, \mathbf{Y}, \boldsymbol{\omega}). \tag{3}$$

We set $q_{\mathbf{M}}(\boldsymbol{\omega}|\mathbf{X}, \mathbf{Y})$ to dropout approximation [7] with variational parameter $\mathbf{M}$. [7] showed that a neural network with Gaussian prior on its weight matrices can be approximated with variational inference, in the form of dropout sampling of deterministic weight matrices and $\ell_2$ weight decay. For the second term, we drop the dependency on $\mathbf{Y}$ (since it is not available in test time) and simply set $q(\mathbf{Z}|\mathbf{X}, \mathbf{Y}, \boldsymbol{\omega})$ to be equivalent to $p_\theta(\mathbf{Z}|\mathbf{X}, \boldsymbol{\omega})$, which works well in practice [23, 28].

Under the SGVB framework [16], we maximize the evidence lower bound (ELBO):

$$\log p(\mathbf{Y}|\mathbf{X}) \geq \mathbb{E}_{\boldsymbol{\omega} \sim q_{\mathbf{M}}(\boldsymbol{\omega}|\mathbf{X}, \mathbf{Y}), \mathbf{Z} \sim p_\theta(\mathbf{Z}|\mathbf{X}, \boldsymbol{\omega})} \left[ \log p(\mathbf{Y}|\mathbf{X}, \mathbf{Z}, \boldsymbol{\omega}) \right] \tag{4}$$

$$- \operatorname{KL}[q_{\mathbf{M}}(\boldsymbol{\omega}|\mathbf{X}, \mathbf{Y})\|p(\boldsymbol{\omega})] - \operatorname{KL}[q(\mathbf{Z}|\mathbf{X}, \mathbf{Y}, \boldsymbol{\omega})\|p_\theta(\mathbf{Z}|\mathbf{X}, \boldsymbol{\omega})] \tag{5}$$

where we approximate the expectation in (4) via Monte-Carlo sampling. The first KL term nicely reduces to $\ell_2$ regularization for $\mathbf{M}$ with dropout approximation [7]. The second KL term vanishes as the two distributions are equivalent. Consequently, our final maximization objective is:

$$\mathcal{L}(\theta, \mathbf{M}; \mathbf{X}, \mathbf{Y}) = \sum \log p_\theta(\mathbf{y}^{(n)}|\tilde{\mathbf{z}}^{(n)}, \mathbf{x}^{(n)}) - \lambda\|\mathbf{M}\|^2 \tag{6}$$

where we first sample random weights with dropout masks $\widetilde{\omega} \sim q_{\mathrm{M}}(\omega|\mathbf{X}, \mathbf{Y})$ and sample $\mathbf{z}$ such that $\tilde{\mathbf{z}} = g(\mathbf{x}, \tilde{\varepsilon}, \widetilde{\omega}), \tilde{\varepsilon} \sim \mathcal{N}(\mathbf{0}, \mathbf{I})$, with a pathwise derivative function $g$ for reparameterization trick. $\lambda$ is a tunable hyperparameter; however in practice it can be simply set to common $\ell_2$ decay shared throughout the network, including other deterministic weights.

When testing with a novel input instance $\mathbf{x}^*$, we can compute the probability of having the correct label $y^*$ by our model, $p(\mathbf{y}^*|\mathbf{x}^*)$ with Monte-Carlo sampling:

$$p(\mathbf{y}^*|\mathbf{x}^*) = \iint p(\mathbf{y}^*|\mathbf{x}^*, \mathbf{z})p(\mathbf{z}|\mathbf{x}^*, \omega)p(\omega|\mathbf{X}, \mathbf{Y})\mathrm{d}\omega\mathrm{d}\mathbf{z} \approx \frac{1}{S}\sum_{s=1}^{S}p(\mathbf{y}^*|\mathbf{x}^*, \tilde{\mathbf{z}}^{(s)}) \qquad (7)$$

where we first sample dropout masks $\widetilde{\omega}^{(s)} \sim q_{\mathrm{M}}(\omega|\mathbf{X}, \mathbf{Y})$ and then sample $\tilde{\mathbf{z}}^{(s)} \sim p_\theta(\mathbf{z}|\mathbf{x}^*, \widetilde{\omega}^{(s)})$.

**Uncertainty Calibration**  The quality of uncertainty from (7) can be evaluated with reliability diagram shown in Figure 1. Better calibrated uncertainties produce smaller *gaps* beween model confidences and actual accuracies, shown in green bars. Thus, the perfect calibration occurs when the confidences exactly matches the actual accuracies: $p(\text{correct}|\text{confidence} = \rho) = \rho, \forall\rho \in [0, 1]$ [9]. Also, [22, 9] proposed a summary statistic for calibration, called the Expected Calibration Error (ECE). It is the expected *gap* w.r.t. the distribution of model confidence (or frequency of bins):

$$\text{ECE} = \mathbb{E}_{\text{confidence}}\big[|p(\text{correct}|\text{confidence}) - \text{confidence}|\big] \qquad (8)$$

## 4  Application to RNNs for Prediction on Time-Series Data

Our variational attention model is generic and can be applied to any generic deep neural network that leverages attention mechanism. However, in this section, we describe its application to prediction from time-series data, since our target application is risk analysis from electronic health records.

**Review of the RETAIN model**  As a base deep network for learning from time-series data, we consider RETAIN [3], which is an attentional RNN model with two types of attentions–across *timesteps* and across *features*. RETAIN obtains state-of-the-art performance on risk prediction tasks from electronic health records, and is able to provide useful interpretations via learned attentions.

We now briefly review the overall structure of RETAIN. We match the notation with those in the original paper for clear reference. Suppose we are interested in a timestep $i$. With the input embeddings $\mathbf{v}_1, \ldots, \mathbf{v}_i$, we generate two different attentions: across timesteps ($\alpha$) and features ($\boldsymbol{\beta}$).

$$\mathbf{g}_i, ..., \mathbf{g}_1 = \text{RNN}_\alpha(\mathbf{v}_i, ..., \mathbf{v}_1; \omega), \qquad \mathbf{h}_i, ..., \mathbf{h}_1 = \text{RNN}_{\boldsymbol{\beta}}(\mathbf{v}_i, ..., \mathbf{v}_1; \omega), \qquad (9)$$

$$e_j = \mathbf{w}_\alpha^\mathsf{T}\mathbf{g}_j + b_\alpha \ \text{ for } \ j = 1, ..., i, \qquad \mathbf{d}_j = \mathbf{W}_{\boldsymbol{\beta}}\mathbf{h}_j + \mathbf{b}_{\boldsymbol{\beta}} \ \text{ for } \ j = 1, ..., i, \qquad (10)$$

$$\alpha_1, ..., \alpha_i = \text{Softmax}(e_1, ..., e_i), \qquad \boldsymbol{\beta}_j = \tanh(\mathbf{d}_j) \ \text{ for } \ j = 1, ..., i. \qquad (11)$$

The parameters of two RNNs are collected as $\omega$. From the RNN outputs $\mathbf{g}$ and $\mathbf{h}$, the attention logits $e$ and $\mathbf{d}$ are generated, followed by squashing functions $\text{Softmax}$ and $\tanh$ respectively. Then the generated two attentions $\alpha$ and $\boldsymbol{\beta}$ are multiplied back to the input embedding $\mathbf{v}$, followed by a convex sum $\mathbf{c}$ up to timestep $i$: $\mathbf{c}_i = \sum_{j=1}^{i} \alpha_j \boldsymbol{\beta}_j \odot \mathbf{v}_j$. A final linear predictor is learned based on it: $\widehat{y}_i = \text{Sigmoid}(\mathbf{w}^\mathsf{T}\mathbf{c}_i + b)$.

The most important feature of RETAIN is that it allows us to interpret what the model has learned as follows. What we are interested in is *contribution*, which shows $x_k$'s aggregate effect to the final prediction at time $j$. Since RETAIN has attentions on both timesteps ($\alpha_j$) and features ($\boldsymbol{\beta}_j$), the computation of aggregate contribution takes both of them into consideration when computing the final contribution of an input data point at a specific timestep: $\omega(y, x_{j,k}) = \alpha_j \mathbf{w}^\mathsf{T}(\boldsymbol{\beta}_j \odot \mathbf{W}_{emb}[:, k])x_{j,k}$. In other words, it is a certain portion of logit $\text{Sigmoid}^{-1}(\widehat{y}_i) = \mathbf{w}^\mathsf{T}\mathbf{c}_i + b$ for which $x_{j,k}$ is responsible.

**Interpretation as a probabilistic model**  The interpretation of RETAIN as a probabilistic model is quite straightforwrad. First, the RNN parameters $\omega$ (9) as gaussian latent variables (1) are approximated with MC dropout with fixed probabilities [7, 5, 27]. The input dependent latent variables $\mathbf{Z}$ (1) simply correspond to the collection of $e$ and $\mathbf{d}$ (10), the attention logits. The log variances of $e$ and $\mathbf{d}$ are generated in the same way as their mean, from the output of RNNs $\mathbf{g}$ and $\mathbf{d}$

| | PhysioNet | | | | Pancreatic | MIMIC |
|---|---|---|---|---|---|---|
| | Mortality | Stay < 3 | Cardiac | Recovery | Cancer | Sepsis |
| RETAIN-DA [3] | 0.7652± 0.02 | 0.8515± 0.02 | 0.9485± 0.01 | 0.8830± 0.01 | 0.8528± 0.01 | 0.7965± 0.01 |
| RETAIN-SA [28] | 0.7635± 0.02 | 0.8412± 0.02 | 0.9360± 0.01 | 0.8582± 0.02 | 0.8444± 0.01 | 0.7695± 0.02 |
| UA-Independent | 0.7764± 0.01 | 0.8572± 0.02 | 0.9516± 0.01 | 0.8895± 0.01 | 0.8533± 0.03 | 0.8019± 0.01 |
| UA | **0.7827± 0.02** | **0.8628± 0.02** | 0.9563± 0.01 | 0.9049± 0.01 | 0.8604± 0.01 | 0.8017± 0.01 |
| UA+ | 0.7770± 0.02 | 0.8577± 0.01 | **0.9612± 0.01** | **0.9074± 0.01** | **0.8638±0.02** | **0.8114± 0.01** |

Table 1: The multi-class classification performance on the three electronic health records datasets. The reported numbers are mean AUROC and standard errors for 95% confidence interval over five random splits.

but with different set of parameters. Also the reparameterization trick for diagonal gaussian is simple [16]. We now maximize the ELBO (6), equipped with all the components $\mathbf{X}, \mathbf{Y}, \mathbf{Z}$, and $\omega$ as in the previous section.

## 5 Experiments

**Tasks and Datasets** We validate the performance of our model on various risk prediction tasks from multiple EHR datasets, for both the prediction accuracy and prediction reliability.

**1) PhysioNet** This dataset [11] contains 4,000 medical records from ICU[3]. Each record contains 48 hours of records, with 155 timesteps, each of which contains 36 physiolocial signals including *heart rate*, *repiration rate* and *temperature*. The challenge comes with four binary classification tasks, namely, 1) *Mortality prediction*, 2) *Length-of-stay less than 3 days:* whether the patient will stay in ICU for less than three days, 3) *Cardiac conditon:* whether the patient will have a cardiac condition, and 4) *Recovery from surgery:* whether the patient was recovering from surgery.

**2) Pancreatic Cancer** This dataset is a subset of the EHR database of the National Health Insurance System (NHIS) in South Korea, consisting of anonymized medical check-up records from 2002 to 2013, which includes around 1.5 million records. We extract 3, 699 patient records from this database, among which 1, 233 are patients diagnosed of pancreatic cancer. The task here is to predict the onsets of pancreatic cancer in 2013 using the records from 2002 to 2012 (11 timesteps), that consists of 34 variables regarding general information (e.g., sex, height, past medical history, family history) as well as vital information (e.g., systolic pressure, hemoglobin level, creatinine level) and risk inducing behaviors (e.g., tobacco and alcohol consumption).

**3) MIMIC-Sepsis** This is the subset of the MIMIC III dataset [12] for sepsis prediction, which consists of 58,000 hospital admissions for 38,646 adults over 12 years. We use a subset that consists of 22,395 records of patients over age 15 and stayed in ICUs between 2001 and 2012, among which 2,624 patients are diagnosed of sepsis. We use the data from the first 48 hours after admission (24 timesteps). For features at each timestep, we select 14 sepsis-related variables including arterial blood pressure, heart rate, FiO2, and Glass Coma Score (GCS), following the clinicians' guidelines. We use Sepsis-related Organ Failure Assessment scores (SOFA) to determine the onset of sepsis.

For all datasets, we generates five random splits of training/validation/test with the ratio of $80\% : 10\% : 10\%$. Detailed description of the datasets, network configuration, and hyperparameters are fully described in the **appendix section**.

**Baselines** We now describe our uncertainty-calibrated attention models and relevant baselines.

**1) RETAIN-DA:** The recurrent attention model in [3], which uses deterministic soft attention.
**2) RETAIN-SA:** RETAIN model with the stochastic hard attention proposed by [28], that models the attention weights with multinoulli distribution, which is learned by variational inference.
**3) UA-independent:** The input-independent version of our uncertainty-aware attention model in (2) whose variance is modeled indepently of the input.
**4) UA:** Our input-dependent uncertainty-aware attention model in (1).
**5) UA+:** The same as UA, but with additional modeling of input-adaptive noise at the final prediction as done in [14], to account for output uncertainty as well.

### 5.1 Evaluation of the binary classification performance

We first examine the prediction accuracy of baselines and our models in a standard setting where the model always makes a decision. Table 1 contains the accuracy of baselines and our models measured

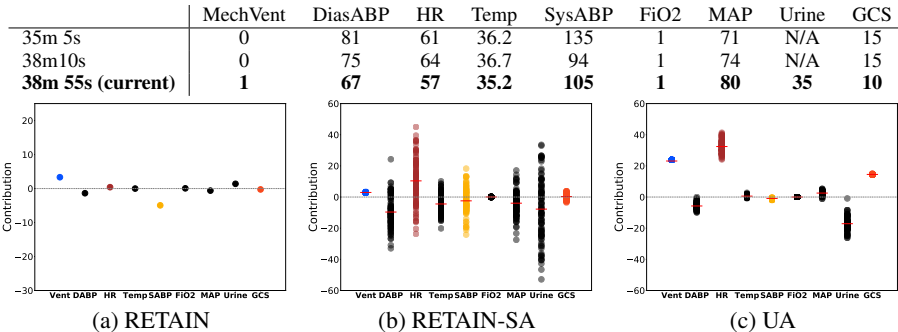

| | MechVent | DiasABP | HR | Temp | SysABP | FiO2 | MAP | Urine | GCS |
|---|---|---|---|---|---|---|---|---|---|
| 35m 5s | 0 | 81 | 61 | 36.2 | 135 | 1 | 71 | N/A | 15 |
| 38m10s | 0 | 75 | 64 | 36.7 | 94 | 1 | 74 | N/A | 15 |
| **38m 55s (current)** | **1** | **67** | **57** | **35.2** | **105** | **1** | **80** | **35** | **10** |

|  (a) RETAIN | (b) RETAIN-SA | (c) UA |
|---|---|---|

Figure 2: Visualization of contributions for a selected patient on PhysioNet mortality prediction task. **MechVent** - Mechanical ventilation, **DiasABP** - Diastolic arterial blood pressure, **HR** - Heart rate, **Temp** - Temperature, **SysABP** - Systolic arterial blood pressure, **FiO2** - Fractional inspired Oxygen, **MAP** - Mean arterial blood pressure, **Urine** - Urine output, **GCS** - Glasgow coma score. The table presents the value of physiological variables at the previous and the current timestep. Dots correspond to sampled attention weights.

in area under the ROC curve (AUROC). We observe that UA variants significantly outperforms both RETAIN variants with either deterministic or stochastic attention mechanisms on all datasets. Note that RETAIN-SA, that generates attention from Bernoulli distribution, performs the worst. This may be because the model is primarily concerned with whether to attend or not to each feature, which makes sense when most features are irrelevant, such as with machine translation, but not in the case of clinical prediction where most of the variables are important. UA-independent performs significantly worse than UA or UA+, which demonstrates the importance of input-dependent modeling of the variance. Additional modeling of output uncertainty with UA+ yields performance gain in most cases.

## 5.2 Interpretability and accuracy of generated attentions

To obtain more insight, we further analyze the contribution of each feature in PhysioNet mortality task in Figure 2 for a patient at the timestep with the highest attention $\alpha$, with the help of a physician. The table in Figure 2 is the value of the variables at the previous checkpoints and the current timestep.

The difference between the current and the previous tmesteps is significant - the patient is applied mechanical ventilation; the body temperature, diastolic arterial blood pressure, and heart rate dropped, and GCS, which is a measure of consciousness, dropped from 15 to 10. The fact that the patient is applied mechanical ventilation, and that the GCS score is lowered, are both very important markers for assessing patient's condition. Our model correctly attends to those two variables, with very low uncertainty. SysABP and DiasABP are variables that has cyclic change in value, and are all within normal range; however RETAIN-DA attended to these variables, perhaps due to having a deterministic model which led it to overfit. Heart rate is out of normal range (60-90), which is problematic but is not definitive, and thus UA attended to it with high variance. RETAIN-SA results in overly incorrect and noisy attention except for FiO2 that did not change its value. Attention on Urine by all models may be the artifact that comes from missing entry in the previous timestep. In this case, UA assigned high variance, which shows that it is uncertain about this prediction.

The previous example shows another advantage of our model: it provides a richer interpretations of why the model has made such predictions, compared to ones provided by deterministic or stochastic model without input-dependent modeling of uncertainty. We further compared UA against RETAIN-DA for accuracy of the attentions, using variables selected meaningful by the clinicians as ground truth labels (avg. 132 variables per

| | Sensitivity | Specificity |
|---|---|---|
| DA | 75% | 68% |
| UA | **87%** | **82%** |

Table 2: Percentage of features selected from each model that match the features selected by the clinicians.

record), from EHRs for a male and a female patient randomly selected from 10 age groups (40s-80s), on PhysioNet-Mortality. We observe that UA generates accurate interpretations that better comply with clinicians' intepretations (Table 2).

## 5.3 Evaluation of prediction reliability

Another important goal that we aimed to achieve with the modeling of uncertainty in the attention is achieving high reliability in prediction. Prediction reliability is orthogonal to prediction accuracy,

| | PhysioNet | | | | Pancreatic | MIMIC |
| | Mortality | Stay $< 3$ | Cardiac | Recovery | Cancer | Sepsis |
|---|---|---|---|---|---|---|
| RETAIN-DA [3] | $7.23 \pm 0.56$ | $2.04 \pm 0.56$ | $5.70 \pm 1.56$ | $4.89 \pm 0.97$ | $5.45 \pm 0.79$ | $3.05 \pm 0.56$ |
| RETAIN-SA [28] | $7.70 \pm 0.60$ | $3.77 \pm 0.07$ | $8.82 \pm 0.64$ | $5.39 \pm 0.80$ | $9.69 \pm 3.90$ | $5.75 \pm 0.29$ |
| UA-Independent | $5.03 \pm 0.94$ | $2.74 \pm 1.44$ | $3.55 \pm 0.56$ | $4.87 \pm 1.46$ | $4.51 \pm 0.72$ | $2.04 \pm 0.62$ |
| UA | $\mathbf{4.22 \pm 0.82}$ | $\mathbf{1.43 \pm 0.53}$ | $3.33 \pm 0.96$ | $4.46 \pm 0.73$ | $3.61 \pm 0.55$ | $\mathbf{1.78 \pm 0.41}$ |
| UA+ | $4.41 \pm 0.52$ | $1.68 \pm 0.16$ | $\mathbf{2.66 \pm 0.16}$ | $\mathbf{3.98 \pm 0.59}$ | $\mathbf{3.22 \pm 0.69}$ | $2.04 \pm 0.62$ |

Table 3: Mean Expected Calibration Error (ECE) of various attention models over 5 random splits.

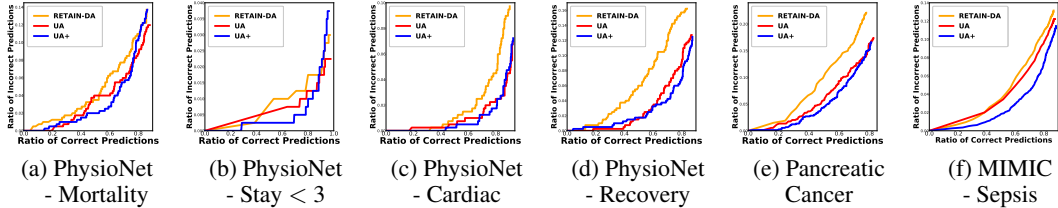

| (a) PhysioNet - Mortality | (b) PhysioNet - Stay $< 3$ | (c) PhysioNet - Cardiac | (d) PhysioNet - Recovery | (e) Pancreatic Cancer | (f) MIMIC - Sepsis |

Figure 3: Experiments on prediction reliability. The line charts show the ratio of incorrect predictions as a function of the ratio of correct predictions for all datasets.

and [22] showed that state-of-the-art deep networks are not reliable as they are not well-calibrated to correlate model confidence with model strength. Thus, to demonstrate the reliability of our uncertainty-aware attention, we evaluate it for the uncertainty calibration performance against baseline attention models in Table 3, using Expected Calibration Errors (ECE) [22] (Eq. (8)). UA and UA+ are significantly better calibrated than RETAIN-DA, RETAIN-SA as well as UA-independent, which shows that independent modeling of variance is essential in obtaining well-calibrated uncertainties.

**Prediction with "I don't know" option**   We further evaluate the reliability of our predictive model by allowing it to say I don't know (IDK), where the model can refrain from making a hard decision of yes or no when it is uncertain about its prediction. This ability to defer decision is crucial for predictive tasks in clinical environments, since those deferred patient records could be given a second round examination by human clinicians to ensure safety in its decision. To this end, we measure the uncertainty of each prediction by sampling the variance of the prediction using both MC-dropout and stochastic Gaussian noise over 30 runs, and simply predict the label for the instances with standard deviation larger than some set threshold as IDK.

Note that we use RETAIN-DA with MC-Dropout [5] as our baseline for this experiment, since RETAIN-DA is deterministic and cannot output uncertainty [4] We report the performance of RETAIN + DA, UA, and UA+ for all tasks by plotting the ratio of incorrect predictions as a function of the ratio of correct predictions, by varying the threshold on the model confidence (See Figure 3). We observe that both UA and UA+ output much smaller ratio of incorrect predictions at the same ratio of correct predictions compared to RETAIN + DA, by saying IDK on uncertain inputs. This suggests that our models are relatively more reliable and safer to use when making decisions for prediction tasks where incorrect predictions can lead to fatal consequences.

## 6   Conclusion

We proposed uncertainty-aware attention (UA) mechanism that can enhance reliability of both interpretations and predictions of general deep neural networks. Specifically, UA generates attention weights following Gaussian distribution with learned mean and variance, that are decoupled and trained in input-adaptive manner. This input-adaptive noise modeling allows to capture heteroscedastic uncertainty, or the instance-specific uncertainty, which in turn yields more accurate calibration of prediction uncertainty. We trained it using variational inference and validated it on seven different tasks from three electronic health records, on which it significantly outperformed the baselines and provided more accurate and richer interpretations. Further analysis of prediction reliability shows that our model is accurately calibrated and thus can defer predictions when making prediction with "I don't know" option.

## Acknowledgments

This work was supported by a Machine Learning and Statistical Inference Framework for Explainable Artificial Intelligence (No.2017-0-01779) funded by Institution for Information & Communications & Technology Promotion (IITP) and Basic Science Research Program through the National Research Foundation of Korea (NRF) funded by the Ministry of Education (2015R1D1A1A01061019) of South Korea. Juho Lee is funded by the European Research Council under the European Union's Seventh Framework Programme (FP7/2007-2013) ERC grant agreement no. 617071.

## Footnotes

[2]The source codes are publicly available at \texttt{https://github.com/jayheo/UA}.

[3]We only use the TrainingSetA, for which the labels were available

[4]RETAIN-SA is not compared since it largely underperforms all others and is not a meaningful baseline.

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
