[Supplementary Material]

# Supplementary File for NIPS 2018 Paper 503: Uncertainty-Aware Attention for Reliable Interpretation and Prediction

**Jay Heo**[1,2,4]*, **Hae Beom Lee**[1,2]*, **Saehoon Kim**[2], **Juho Lee**[2,5], **Kwang Joon Kim**[3], **Eunho Yang**[1,2], **Sung Ju Hwang**[1,2]

KAIST[1], AItrics[2], Yonsei University College of Medicine[3], UNIST[4], South Korea,
University of Oxford[5], United Kingdom,
{sflame87, haebeom.lee, sjhwang82, eunhoy}@kaist.ac.kr
shkim@aitrics.com, preppie@yuhs.ac, juho.lee@stats.ox.ac.uk

## 1 Application to Convolutional Neural Networks for Image Classification

To further show the generality of our uncertainty-aware attention mechanism, we performed additional experiments of our model with convolutional neural networks on image classification tasks.

**Approach** Given a convolutional layer $\mathbf{h} \in \mathbb{R}^{K \times H \times W}$ where $K$ is the number of channels, and $H$ and $W$ are height and width of the feature map, we can generate from $\mathbf{h}$ an attention map for spatial grids and its variance: $\boldsymbol{\mu}(\mathbf{h}), \boldsymbol{\sigma}(\mathbf{h}) \in \mathbb{R}^{1 \times H \times W}$. (For example, we used three $3 \times 3$ convolutional layers with ReLU nonlinearity, where the last convolutional layer is splitted into two layers to generate $\boldsymbol{\mu}$ and $\boldsymbol{\sigma}$, respectively). Softplus function is used for $\boldsymbol{\sigma}$ for nonnegativity. We apply dropout before each convolutional layer in generating $\boldsymbol{\mu}$ and $\boldsymbol{\sigma}$ in order to respresent model uncertainty, following Gal and Ghahramani [1]. We then combine $\boldsymbol{\mu}$ and $\boldsymbol{\sigma}$ into Gaussian attention logit $\mathbf{z}$.

$$\mathbf{z} \sim \mathcal{N}(\boldsymbol{\mu}(\mathbf{h}), \operatorname{diag}(\boldsymbol{\sigma}(\mathbf{h})^2)) \tag{1}$$

We can optionally squash it as $\boldsymbol{\alpha} = \operatorname{Sigmoid}(\mathbf{z})$; however for this experiment we did not apply squashing function for our model ($\boldsymbol{\alpha} = \mathbf{z}$). The attention $\boldsymbol{\alpha} \in \mathbb{R}^{1 \times H \times W}$ is then multiplied back to $\mathbf{h} \in \mathbb{R}^{K \times H \times W}$, that is, $\mathbf{h}_{att} = \mathbf{h} \odot \boldsymbol{\alpha} \in \mathbb{R}^{K \times H \times W}$, where $\odot$ is an element-wise multiplication operation w.r.t. height and width.

**Dataset and baselines** For dataset, we use two versions of MNIST-Variation [5] dataset to see if our model helps improve classification performance on such noisy samples with large appearance ambiguities. The first dataset is MNIST-Rotation, which contains rotated MNIST digits. The second one is MNIST-Rotation & Background dataset, which is further applied with random backgrounds. For each dataset, we use 1,000 and 50,000 instances for training and test respectively. We apply our attention mechanism to the first convolutional layer of LeNet [4] which has $12 \times 12$ spatial grids and compare this modified model against the same base network with various types of attention mechanisms:

**1) LeNet:** Base convolutional neural networks [4] with 2 Conv and 2 FC layers.
**2) LeNet-DA:** LeNet with deterministic soft attention on the first convolutional layer (Conv1); attentions are generated with sigmoid output units and placed on spatial grids.
**3) LeNet-SA:** The same as LeNet-DA, but with the deterministic attentions replaced with stochastic Bernoulli variables sampled from the learned attention probabilities.
**4) UA:** LeNet augmented with our uncertainty-aware attention mechanism. Note that we apply dropout at convolutional layers with $0.05$ retain probability.

---

| | MNIST Rotation | | MNIST Rotation & Background | |
|---|---|---|---|---|
| | Accuracy (%) | ECE | Accuracy (%) | ECE |
| LeNet | $76.92 \pm 0.34$ | $2.55 \pm 0.05$ | $47.82 \pm 0.20$ | $7.12 \pm 0.03$ |
| LeNet-DA | $76.80 \pm 0.17$ | $2.61 \pm 0.04$ | $47.86 \pm 0.18$ | $7.09 \pm 0.04$ |
| LeNet-SA | $79.89 \pm 0.43$ | $1.48 \pm 0.04$ | $50.61 \pm 0.35$ | $5.00 \pm 0.08$ |
| UA | $\mathbf{82.57 \pm 0.33}$ | $\mathbf{1.32 \pm 0.08}$ | $\mathbf{52.58 \pm 0.13}$ | $\mathbf{4.68 \pm 0.80}$ |

Table 1: **Results from the MNIST-Variation experiment with LeNet.** The reported numbers are mean classification accuracies and expected calibration error (ECE) scores over 5 runs.

Figure 1: The visualization of the learned attention $\boldsymbol{\alpha}$, in terms of its strength $\mathbb{E}[\boldsymbol{\alpha}]$ and uncertainty $\text{std}(\boldsymbol{\alpha})$ from the MNIST Rotation & Background dataset.

**Experimental Setup**  For all the baselines and our model, we used Adam optimizer [3] and train for 200 epochs with batch size 100. Learning rate is $10^{-3}$ and multipled by 0.1 at $\frac{1}{3}$ and $\frac{2}{3}$ of total epochs. Weight decay is set to $10^{-3}$.

**Analysis**  The results in Table 1 show that the deterministic soft attention (LeNet-DA) does not improve the performance of the base network[2]. However, SA outperforms base network in both the accuracy and reliability. Such good performance of SA mostly comes from its ability to generate sharper attention focusing on the spatial bins that actually contain the digits, compared to DA that generates overly smooth attentions. Yet, our UA model even outperforms this LeNet-SA on both accuracy and expected calibration error (ECE). To understand where the performance inprovements of our model come from, we visualize the learned attention strength and variance for some example images in Figure 1.

We explain the source of improvements as follows. Firstly, we can see from the easy examples (Left) that the SA represents high uncertainty on the black monotonous backgrounds, which is not reasonable as the most confusions may occur on the center digits for these easy examples. This is because while the mean of bernoulli variable $\mathbb{E}[\boldsymbol{\alpha}] = \mathbf{p}$ tends to focus on the center, the variance $\text{std}(\boldsymbol{\alpha}) = \sqrt{\mathbf{p} \odot (1 - \mathbf{p})}$ is negatively correlated with it when $p_i > 0.5$ (for each location $i$) and hence focus on the corners. On the other hand, our UA model effectively learns higher variance $\text{std}(\boldsymbol{\alpha}) = \boldsymbol{\sigma}$ on the center by disentangling the variance from the mean $\mathbb{E}[\boldsymbol{\alpha}] = \boldsymbol{\mu}$.

Secondly, for DA and SA model, the learned attention strength for hard examples (Right) is stronger than that of easy examples. This is because those models are not aware of epistemic uncertainty that comes from the lack of data. Our model can effectively capture this model uncertainty in the form of dropout noise [2, 1], and puts weaker attention strength on these hard examples, effectively preventing overfitting.

## 2 Detailed Description of Datasets and Experimental Setup

### 2.1 Datasets

**MIMIC3-Sepsis**  We calculated Sepsis-related Organ Failure Assessment Score(SOFA) [6] for each patient to determine the onset of sepsis: if SOFA score increases by 2 points or more within the time window, we label the patient as positive. We set the time window as 72 hours, since the current guideline of American Medical Association considers the specified period of suspected infection on sepsis as 48 hours before and up to 24 hours after the onset of sepsis [6]. The overal rate of septic patients is 16.07%. Table 4 describes feature information in details. We selected features under

the guidelines of physicians and, for urine outputs, we adopted the similar approach to the recent work [7]: we sum the variables representing urine.

**Pancreatic Cancer**   This datasets is a subset of electronic healthcare records-based database from healthcare organization, consisting of around 1.5 million records. The database contains demographic information including medical aid beneficiaries, treatmenet information, disease histories, and drug prescription records. In total, 34 features regarding vital signs, social and behavioral factors, medical history, and general information, were extracted from the database over 12 years. Total cholesterol level and fasting glucose levle were sampled after overnight fasting and systolic blood pressure and diastolic blood pressure were checked through medical examinations. Also, there were several questionnaires that are designed to identify social and behavioral risk factors, such as smoking habit, alcohol consumption, and time spent on excercise. Individual medical history was followed with drug perscription history and clinical codes of the 10th revision of the International Classification of Diseases (ICD-10). We determined patients with pancreatic cancer by identifying ICD code, C25, on examination and treatment records. On the labeling process, we exclude those who had previous pancreatic cancer-related treatment records as well as pre-existing medical history of pancreatic cancer. Table 5 describes feature information in details.

## 2.2   Configuration and Parameters

We trained all the models using Adam [3] optimizer with dropout regularization. We set the maximum iteration for Adam optimizer as $100,000$, and for other hyperparameters, we searched for the optimal values by cross-validation, within predefined ranges as follows: Mini batch size: $\{32, 64, 128, 256\}$, learning rate: $\{0.01, 0.001, 0.0001\}$, *L-2* regularization: $\{0.02, 0.002, 0.0002, 0.0004\}$, and dropout rate $\{0.1, 0.15, 0.2, 0.25, 0.3, 0.4, 0.5\}$.

## 3   Benefits of Input-adaptive Uncertainty Modeling

We conducted experiments to show the benefits of input-adaptive noise on PhysioNet-Mortality dataset. First, we intentionally corrupted the distribution of original dataset with Gaussian noise. The result shows that UA and UA+ outperform RETAIN in classification performance. Especially, when comparing measured attention weights on noisy features, UA captures $86\%$ of noisy features, while RETAIN captures only $59\%$ with a threshold of attention weight, $0.01$. For the second experiment, we intentionally increased the original missing rate by $5\%$, from $92\%$ to $97\%$, to simulate low-quality samples. As a result, UA and UA+ models outperform RETAIN in classification performance.

|  | Gaussian Noise | 97% Missing Rate |
| --- | --- | --- |
| RETAIN-DA | 0.7692 | 0.7129 |
| UA | **0.7868** | 0.7372 |
| UA+ | 0.7864 | **0.7643** |

Table 2: Classification performance of RETAIN and uncertainty-aware attention models on PhysioNet-Mortality dataset. The reported numbers are AUROC.

## 4   Prediction with "I Don't Know" Decision

We analyzed the predictions for PhysioNet-Mortality to address how many of the IDK predictions would have been false positives, false negatives, or true positives. The result shows that, when correct prediction rate becomes 0.7, UA mainly filters out more false negative cases, while RETAIN filters out more false positive cases. This is a promising result since preventing type II error is critical for healthcare applications.

In Figure 3, we observe that both UA and UA+ are more likely to say IDK rather than make incorrect predictions when compared against RETAIN + MC Dropout model, which suggests that they are relatively more reliable, and safer to use for making clinical decisions where incorrect predictions can lead to fatal consequences. For instance, on sepsis prediction task, UA+ made

|  | (a)RETAIN | (b)UA | (c)UA+ |

Figure 2: Uncertainty over prediction strength on PhysioNet Challenge dataset. For all models, we measured the prediction uncertainty by using MC-dropout with 30 samples.

|  | False Positive | False Negative | True Positive |
|---|---|---|---|
| RETAIN-DA | **14** | 14 | 8 |
| UA | 7 | **22** | **10** |
| UA+ | 8 | 21 | 9 |

Table 3: Number of false positives, false negatives, and true positives in IDK holder on PhysioNet-Mortality dataset.

incorrect prediction only on $0.17\%$ of the instances ($0.80\%$ for UA) while avoiding $29.83\%$ of potentially incorrect predictions based on uncertainty, when correct prediction rate becomes 0.7. On the other hand, RETAIN + MC Dropout predicted incorrectly on $2.51\%$ of the instances with $27.68\%$ IDK predictions. Considering the consequences that follow an incorrect prediction of sepsis, this is a significant difference. Furthermore, for pancreatic cancer prediction task, our model made $14.32\%$ incorrect predictions with $15.68\%$ IDK decisions, while RETAIN + MC Dropout made incorrect prediction on $17.54\%$ of instances with $12.46\%$ IDK decisions. This difference is significant considering the severe consequences an incorrect cancer prediction has on the patient.

## Footnotes

[2]We also tried a DA model without squashing but it performed similar to DA.

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

(a) PhysioNet-Mortality    (b) PhysioNet-Stay    (c) PhysioNet-Cardiac

(d) PhysioNet-Recovery    (e) Pancreatic Cancer    (f) MIMIC-Sepsis

Figure 3: Experiments on prediction reliability. The stacked bar charts show the ratio of IDK and incorrect predictions, when correct prediction becomes 0.7.

Table 4: Feature information of MIMIC-Sepsis dataset.

| Features | Item-ID | Name of Item |
|---|---|---|
| Age | N/A | intime<br>dob |
| Heart rate | 211<br>22045 | Heart Rate<br>Heart Rate |
| FiO2 | 223835<br>3420<br>3422<br>190 | Inspired O2 Fraction<br>FiO2<br>FiO2 [Meas]<br>FiO2 set |
| Temperature | 676<br>678<br>223761<br>223762 | Temperature C<br>Temperature F<br>Temperature Fahrenheit<br>Temperature Celsius |
| Systolic Blood Pressure | 51<br>442<br>455<br>6701<br>220179<br>220050 | Arterial BP[Systolic]<br>Manual BP[Systolic]<br>NBP[Systolic]<br>Arterial BP #2 [Systolic]<br>Non Invasive Blood Pressure[systolic]<br>Arterial Blood Pressure[systolic] |
| Diastolic Blood Pressure | 8368<br>8440<br>8441<br>8555<br>220051<br>220180 | Arterial BP[Diastolic]<br>Manual BP[Diastolic]<br>NBP[Diastolic]<br>Arterial BP #2[Diastolic]<br>Non Invasive Blood Pressure[Diastolic]<br>Arterial Blood Pressure[Diastolic] |
| PaO2 | 50821<br>50816 | PO2<br>Oxygen |
| GCS - Verbal Response | 223900 | Verbal Response |
| GCS - Motor Response | 223901 | Motor Response |
| GCS - Eye Opening | 220739 | Eye Opening |
| Serum Urea Nitrogen Level | 51006 | Urea Nitrogen |
| Sodium Level | 950824 | Sodium Whole Blood |
| White Blood Cells Count | 51300<br>51301 | WBC Count<br>White Blood Cells |
| Urine Output | 40055<br>43175<br>40069<br>40094<br>40715<br>40473<br>40085<br>40057<br>40056<br>40405<br>40428<br>40086<br>40096<br>40651<br>226559<br>226560<br>226561<br>226584<br>226563<br>226564<br>226565<br>226567<br>226557<br>226558<br>227488<br>227489 | Urine Out Foley<br>Urine<br>Urine Out Void<br>Urine Out Condom Cath<br>Urine Out Suprapubic<br>Urine Out IleoConduit<br>Urine Out Incontinent<br>Urine Out Rt Nephrostomy<br>Urine Out Lt Nephrostomy<br>Urine Out Other<br>Urine Out Straight Cath<br>Orine Out Incontinent<br>Urine Out Ureteral Stent #1<br>Urine Out Ureteral Stent #2<br>Foley<br>Void<br>Condom Cath<br>Ileoconduit<br>Suprapubic<br>R Nephrostomy<br>L Nephrostomy<br>Straight Cath<br>R Ureteral Stent<br>L Ureteral Stent<br>GU Irrigant Volumne In<br>GU Irrigant/Urine Volume Out |

Table 5: Feature information of pancreatic cancer dataset.

| Category | Feature |
|---|---|
| Demographics | Age |
| | Sex |
| Socio-Economic Status | Income Level |
| | Type of Disability |
| Health Screening | Body Mass Index (BMI) |
| | Waist Circumference |
| | Systolic Blood Pressure |
| | Diastolic Blood Pressure |
| | Fasting Glucose |
| | Total Cholesterol |
| | Triglyceride |
| | Hemoglobin |
| | Urine Protein |
| | Creatinine |
| | HDL Cholesterol |
| | LDL Cholesterol |
| | Aspartate Aminotransferase |
| | Alanine Transaminase |
| | Gamma-Glutamyl Transferase |
| Family History | Liver Disease |
| | Stroke |
| | Heart Disease |
| | Hypertension |
| | Diabetes Mellitus |
| | Cancer |
| Personal History | Stroke or Cerebral Infarction-related Disease |
| | Heart Disease |
| | Hypertension |
| | Diabetes Mellitus |
| | Hyperlipidemia |
| | Tuberculosis |
| Social and behavioral Factor | Alcohol Consumption |
| | Smoking Habit |
| | Physical Exercise |