[Reviews · NeurIPS 2018]

Reviewer 1



The paper was well written and easy to read. The authors propose "uncertainty-aware" attention, that models attention weights as Gaussian with the variance being a function of the input. Strengths: - the uncertainty-aware variational approach is novel and can be used for a variety of tasks - the resulting attentions have greater interpretability (table 2) and prediction reliability (figure 3) Weaknesses: - It would have been interesting to see experimental results on a different task Question for authors: Do the authors have an intuition for the approximately 13% error in interpretability?

Reviewer 2



This paper addresses measuring the uncertainty of model predictions and evaluating the importance of input features. The work proposed to apply dropout and l2 weight decay [7] on the model, and add input dependent noise on stochastic attention model. The evaluation compared the results on electronic health record datasets with conventional approaches that use soft attention and stochastic hard attention (which use input-independent noise). The work also evaluated the quality of feature selection by the attention mechanism through comparing it with human selected features, and shown improvement compared to the soft attention. The authors show that the resulting uncertainty prediction allows their models to filter out more false negative. Being able to provide uncertainty of model prediction is a critical criterion for medical domain, and the proposed approach provide reasonable improvement over existing approaches. On the other hand, the novel part of this paper is to combine [7] with stochastic attention mechanism and proposed input dependent noise, which has some but limited originality. The paper is clearly written.

Reviewer 3



The paper offers a method to estimate data-dependent uncertainty in attention mechanisms in current neural networks using a Bayesian setting. Applications to a particular RNN model known as RETAIN, running on three time-series health data show promising results. Pros: - The problem under investigation is important, especially in critical domains such as healthcare - The approach through Bayesian inference with variational approximation is reasonable Cons: - The novelty is not very significant as it is an incremental step from current advances in Bayesian deep learning - Evaluation on RETAIN only is somewhat limited, as there is no reason to stick with just one particular RNN variant. For health time-series, a major problem is irregular in data sampling, and this confounds with the uncertainty estimate.